# Synthesis of Zn^2+^-Pre-Intercalated V_2_O_5_·nH_2_O/rGO Composite with Boosted Electrochemical Properties for Aqueous Zn-Ion Batteries

**DOI:** 10.3390/molecules27175387

**Published:** 2022-08-24

**Authors:** Yanzhi Fan, Xiaomeng Yu, Ziyi Feng, Mingjie Hu, Yifu Zhang

**Affiliations:** 1Beijing Aerospace Intelligent Construction Co., Ltd., Beijing 102600, China; 2State Key Laboratory of Fine Chemicals, School of Chemical Engineering, Dalian University of Technology, Dalian 116024, China; 3Hubei Key Laboratory of Advanced Aerospace Propulsion Technology, Hubei Military-Civilian Integration and Co-Innovation Center of Aerospace Propulsion and Materials Technology, Wuhan 430040, China

**Keywords:** composite materials, Zn_x_V_2_O_5_·nH_2_O/rGO, electrochemical properties, energy storage and conversion, Zn-ion batteries

## Abstract

Layered vanadium-based materials are considered to be great potential electrode materials for aqueous Zn-ion batteries (AZIBs). The improvement of the electrochemical properties of vanadium-based materials is a hot research topic but still a challenge. Herein, a composite of Zn-ion pre-intercalated V_2_O_5_·nH_2_O combined with reduced graphene oxide (ZnVOH/rGO) is synthesized by a facile hydrothermal method and it shows improved Zn-ion storage. ZnVOH/rGO delivers a capacity of 325 mAh·g^−1^ at 0.1 A·g^−1^, and this value can still reach 210 mAh·g^−1^ after 100 cycles. Additionally, it exhibits 196 mAh·g^−1^ and keeps 161 mAh·g^−1^ after 1200 cycles at 4 A·g^−1^. The achieved performances are much higher than that of ZnVOH and VOH. All results reveal that Zn^2+^ as “pillars” expands the interlayer distance of VOH and facilitates the fast kinetics, and rGO improves the electron flow. They both stabilize the structure and enhance efficient Zn^2+^ migration. All findings demonstrate ZnVOH/rGO’s potential as a perspective cathode material for AZIBs.

## 1. Introduction

Nowadays, environmental problems and the energy crisis have become dominating hot topics. Researchers are engaged in exploring various high-efficiency, green energy storage and conversion devices [1,2,3]. Among various energy storage systems, aqueous Zn-ion batteries (AZIBs) have attracted increasing interest in recent years as an alternative for grid-scale energy storage systems because of their low-cost, eco-friendliness and safety [4,5,6,7]. For AZIBs, metal zinc is typically used as the anode due to its high theoretical capacity of 820 mAh·g^−1^ and abundance. However, a lot of work is still ongoing aimed at finding suitable cathode materials that meet the requirements of high energy density and long-cycle life [8,9,10]. 

So far, various cathode materials for AZIBs, such as manganese oxides, vanadium oxides and their related compounds, Prussian blue analogues, cobalt oxides, organic molecules and so on, have been widely studied [11,12,13,14,15,16]. Among them, vanadium (V)-based compounds are considered as attractive candidates because of their variable valence states, abundant reserves and open-frame structures [17,18,19,20,21]. This material can accommodate lots of Zn-ions. Layered hydration vanadium pentoxide V_2_O_5_·nH_2_O (abbreviation as VOH) possesses tunable structures and multivalences for large Zn^2+^ reserves and ingress/egress [22,23,24,25]. Although many progresses of vanadium oxides are achieved as cathode materials for AZIBs, they still suffer from intrinsic poor electroconductivity and hindered dynamics owing to the strong electrostatic attraction between Zn^2+^ and VO layer [26]. Thus, the structural engineering of vanadium oxides with open architecture is key to eliminating this electrostatic impact [27].

As the cathode material for AZIBs, Zn_0.25_V_2_O_5_∙nH_2_O was first reported by Nazar’s group [28]; it exhibited high capacity (ca. 300 mAh g^−1^ @ 50 mA g^−1^) and showed excellent rate performance. After that, some objects (e.g., various metal ions, different polymers) are introduced to tune the lamellar structure of VOH. After adjusting the interlayer spacing, electrochemical properties of intercalated VOH were significantly boosted [29,30,31,32,33,34]. For instance, Liu et al. reported that polyaniline-intercalated VOH shows an expanded interlayer space of ~14 Å delivering a capacity of 354 mAh·g^−1^ at 0.1 A·g^−1^ and good cycle stability [35]. Other reports prove the utility of object-intercalated VOH with enhanced electrochemical properties for AZIBs. Although Zn_0_._25_V_2_O_5_∙nH_2_O for AZIBs has been reported, Zn^2+^-intercalated VOH combined with rGO has been rarely reported to the best of our knowledge [36]. As is well known, recombination rGO with metal oxides enhances the conductivity, facilitates electron flow and stabilizes the structure of the composites [37,38]. 

Herein, a composite with Zn^2+^-intercalated VOH integrated with rGO (denoted as ZnVOH/rGO) is developed and it exhibits enhanced Zn-ion storage with a capacity of 325 mAh·g^−1^ at 0.1 A·g^−1^ and 210 mAh·g^−1^ after 100 cycles. The capacity is not higher than that reported by Nazar (0.05 A·g^−1^) because of our value obtained at a very high rate 0.1 A·g^−1^. The achieved performances of ZnVOH/rGO are much higher than that of ZnVOH and VOH, and even surpass some state-of-the-art cathodes for the application of AZIBs.

## 2. Results and Discussion

ZnVOH/rGO was prepared by a hydrothermal route combination with freeze-drying treatment. V-based complexes were first formed by reacting between V_2_O_5_ and H_2_O_2_ to form a clear solution [35]. In hydrothermal route, VOH nanobelts were formed, meanwhile, Zn-ions were intercalated into a VOH interlayer, expanding the distance of interlayer spacing, like other metal ions [30]. During this process, GO was transformed to rGO and ZnVOH was connected/fixed with rGO by hydroxyl bonding or electrostatic attraction [38]. ZnVOH/rGO was formed by this hydrothermal route and finally obtained by freeze-drying treatment.

Figure 1 shows the XRD patterns of ZnVOH and ZnVOH/rGO. The characteristic peaks correspond to the standard pattern of V_2_O_5_·1.6H_2_O (JCPDS, card No. 40-1296) [39]. The main peaks of the four materials matched well with PDF#40-1296, but due to the intercalation of metal ions, the peak position of VOH will be shifted. These phenomena are reported in the previous literature [39]. The (001) peak with the lattice spacing of 13.1 Å is achieved from ZnVOH/rGO, which is larger than the value (11.9 Å) of ZnVOH. The increase of interlayer spacing not only expands the transport channel of zinc ions, but also weakens the electrostatic interaction with the main material. In order to determine the composition of the materials, we conducted the ICP test for the two materials. The result is shown in Appendix A. The Zn/V ratios in ZnVOH and ZnVOH/rGO are similar, both about 0.11:1. This is very consistent with the synthesis of ZnVOH/rGO compared with ZnVOH only the addition of rGO. From the test results, it is found that the ratio of Zn to V does not change greatly with the addition of rGO, but the addition of rGO greatly improves the layer spacing. This may be because the introduction of rGO makes more water molecules enter the layer between vanadium oxides. This result suggests the addition of GO can not only facilitate the fast Zn ion transfers but also improve the electrochemical conductivity of ZnVOH [40]. In our experiment, we also investigated the amount of rGO. However, the interlayer spacing of ZnVOH/rGO has changed little with the change of rGO.

Figure 2a depicts FTIR spectra of ZnVOH and ZnVOH/rGO. The characteristic bands at 524, 728 and 836 cm^−1^ are assigned to the stretching vibrations of O-V-O. The bands at 995 and 1015 cm^−1^ are assigned to V^4+^ = O and V^5+^ = O, respectively [41]. Meanwhile, some emerged characteristic bands at 1095, 1147 and 1730 cm^−1^ are observed, which are attributed to the C-H, C-O and C = O bonds from rGO [42]. More structural details of ZnVOH and ZnVOH/rGO are provided by Raman spectra, as depicted in Figure 2b. The Raman peaks below 1000 cm^−1^ are assigned to (V_2_O_2_)n, V_3_-O, V-O-V and V = O from ZnVOH [43]. Figure 2b gives Raman spectra of ZnVOH and ZnVOH/rGO. In ZnVOH/rGO, two characteristic peaks at approximately 1340 and 1580 cm^−1^ observed, which are indexed to D-band and G-band, respectively. These two bands are related to defective carbon atoms (D-band) and ordered carbon atoms (G-band) [38]. The ratio of peak strength of I_D_/I_G_ suggests the carbon defect density [38]. Materials with a larger ratio of I_D_/I_G_ suggest a higher carbon defect density, and a better electrochemical performance [40]. The I_D_/I_G_ value of ZnVOH/rGO is 1.02, which means that the composite has a relative higher carbon defect density. This is conducive to energy storage. Moreover, compared to the ZnVOH, the Raman spectrum of ZnVOH/rGO shows peaks at approximately 2700 cm^−1^, which is attributed to the second-order peak of the D peak (2D-band). This further demonstrates the existence of graphene layers [44]. The above results suggest the successful preparation of the ZnVOH/rGO composite.

The morphologies of the samples were investigated by SEM and TEM. Figure 3 and Appendix A show the morphologies of ZnVOH and ZnVOH/rGO, respectively. Both samples show uniform nanowires. As shown in Figure 3a,b, ZnVOH nanowires tightly stick to rGO sheets through chemical bonds or electrostatic attraction (ZnVOH nanowires were shown in Appendix A). The results suggest that ZnVOH nanowires and rGO sheets are dispersed well, endowing the electrical conductivity of ZnVOH nanowires. Furthermore, the element of ZnVOH/rGO is advocated by elemental mapping images, as shown in Figure 3c and Appendix A. The elements V, O, Zn and C are uniformly dispersed in ZnVOH/rGO, in line with the results of XRD, FTIR and Raman. We tested the quality of the ZnVOH/rGO before and after calcination, and the content of rGO was about 7.1%. Figure 3d–f represents the TEM images of ZnVOH/rGO, which also reveal that ZnVOH nanowires are densely anchored on rGO sheets, where the strong interaction with each other is formed. This architecture can effectively restrain the aggregation of rGO through π stacking interactions [45]. An HRTEM image inserted in Figure 3f reveals that the lattice distance of (001) plane is expanded to 13.1 Å, which is in line with XRD observation (Figure 1). All the above results confirm that ZnVOH/rGO with an expanded layer spacing is successfully synthesized. This feature not only facilitates fast kinetics and efficient ion-transfer, but also enhances structural stability and electrical conductivity [46].

The Zn-ion storage performance of ZnVOH/rGO was tested (details in Appendix A). Figure 4a depicts CV curves of ZnVOH and ZnVOH/rGO at scan rate of 0.1 mV·s^−1^ in 0.2~1.4 V potential range. They have two pairs of redox peaks, and they are assigned to the reactions of V^5+^/V^4+^ and V^4+^/V^3+^, respectively. As for ZnVOH, these two pairs of redox peaks are located at 0.428/0.771 V and 0.760/1.201 V. However, they are shifted to 0.565/0.752 V and 0.843/1.045 V for ZnVOH/rGO. The potential gap of ZnVOH/rGO is much smaller than that of ZnVOH, revealing better redox reaction kinetics and faster ion diffusion leading to the smaller polarization. The above result suggests that the introduction of rGO is good for the redox reaction kinetics, and it also plays a role in promoting thermodynamics [47]. Figure 4b shows the CV curves of ZnVOH/rGO at the initial cycles. These curves are well overlapped, which indicates the excellently reversible Zn^2+^ (de)intercalation of ZnVOH/rGO. Furthermore, two couples of redox peaks in all CV curves (Figure 4) are indexed to ingress/egress of Zn^2+^, suggesting a multistep intercalation mechanism [29].

Figure 5 and Appendix A represent GCD curves for different cycles and cycle performances of ZnVOH and ZnVOH/rGO at 0.1 and 4 A·g^−1^, respectively. There are two voltage plateaus in all GCD curves (Figure 5a and Appendix A), which well coincide to the multistep intercalation process in CV curves. It is noted that in the ZnVOH/rGO composite we regard ZnVOH and rGO as a whole as the active material to calculate the electrochemical capacity. At 0.1 A·g^−1^ (Figure 5b), ZnVOH/rGO exhibits a specific capacity of 325 mAh·g^−1^ and this value still reach 210 mAh·g^−1^ after 100 cycles, whereas ZnVOH displays lower specific capacity and these two values of ZnVOH are 295 and 140 mAh·g^−1^. At 4 A·g^−1^ (Appendix A), ZnVOH/rGO exhibits a capacity of 196 mAh·g^−1^ and it retains 161 mAh·g^−1^ after 1200 cycles, whereas ZnVOH exhibits a capacity of 145 and 126 mAh·g^−1^. Moreover, the coulombic efficiencies of ZnVOH/rGO are near 100%, which means that ZnVOH/rGO is more stable than ZnVOH (Figure 5b and Appendix A). The electrochemical performance of the ZnVOH mixture electrode with the same amount of C (carbon black + C coming from rGO), and its electrochemical performance is far inferior to that of ZnVOH/rGO. The results show that the connection of rGO to ZnVOH is conducive to the rapid transfer of electrons [38].

Figure 6 describes the rate performances of ZnVOH and ZnVOH/rGO at 0.1, 0.2, 0.5, 1 and 2 A·g^−1^ and then returns to 0.1 A·g^−1^. From the Figure 6, it is clearly seen that ZnVOH/rGO shows higher electrochemical properties compared to ZnVOH. In detail, at 0.1, 0.2, 0.5, 1 and 2 A·g^−1^, ZnVOH/rGO delivers capacities of 306, 298, 282, 265 and 246 mAh·g^−1^, respectively. This means, compared with the value at 0.1 A·g^−1^, the rate retention of ZnVOH/rGO is near 80%. For ZnVOH, this rate retention is approximately 48%. After the current density is back to 0.1 A·g^−1^, the capacities of ZnVOH/rGO and ZnVOH are 274 and 229 mAh·g^−1^, respectively, suggesting that these values return to 90% and 86% of the initial values. According to the mass of ZnVOH/rGO, the energy densities of Zn//ZnVOH/rGO cell are calculated to be 249 and 225 Wh·kg^−1^ at 0.1 and 2 A·g^−1^, respectively. The above findings reveal that ZnVOH/rGO exhibits much better electrochemical properties (specific capacity, rate and performances) than ZnVOH and VOH [38]. This shows that rGO enhances the rapid electron transport and boosts the electrical conductivity. Furthermore, the achieved performance of ZnVOH/rGO is superior to some recently reported cathodes for AZIBs, as listed in Appendix A.

To better understand why the Zn//ZnVOH/rGO battery has good electrochemical properties, the electrochemical kinetics of Zn//ZnVOH/rGO cell was further studied by CV curves at the scan rage of 0.2 to 1.0 mV·s^−1^, as shown in Figure 7. The shapes of these shapes of curves are similar and the redox peaks display some shifts and become broader with the scan rate increasing (Figure 7a). The reason for such variation is owing to the polarization effect and can be used to study the detailed kinetic [48]. In general, the empirical power–law relationship between the peak current (*i*) and scanning rates (*v*) can be depicted in Equation (1):*i* = *a**ν**^b^*(1)
*log*(*i*) = *log*(*a*) + *blog*(*υ*)(2)

In these two equations [49,50], *a* and *b* are parameters. *b* often ranges from 0.5 (diffusion behavior) to 1 (capacitive behavior) and is related to the charge storage mechanism. To obtain *b* values, the linear relation between log (*i*) and log (*v*) is calculated by Equation (2), as represented in Figure 7b. The *b* values of these four peaks are fitted to be 0.84, 0.92, 0.89 and 0.93, respectively. These *b* values are near 1, revealing that the capacitive process dominates [22]. 

The current (*i* = *a*υ*^b^*) can be further separated into two parts. The ratio of capacitive behavior is acquired by Equations (3) and (4):*i* = k_1_υ + k_2_υ^1/2^(3)
*i/*υ^1/2^ = k_1_υ^1/2^ + k_2_(4)

In these two equations, k_1_υ represents the capacitive behavior, and k_2_υ^1/2^ means the diffusion-controlled behavior. The contribution of capacitive behavior is quantitatively reflected by current response at typical voltages, and the representative result at 0.2 mV s^−1^ is shown in Figure 7c. The contribution rates of capacitive behavior are measured to be 59%, 62%, 68%, 70% and 75% at 0.2, 0.4, 0.6, 0.8 and 1 mV·s^−1^, respectively. This also reveals the major role of capacitive behavior (Figure 7d).

## 3. Materials and Methods

All materials are presented in the Appendix A and were used as received. The GO solution was synthesized by the reported improved Hummer’s method [24]. To synthesize ZnVOH/rGO, 1 mmol V_2_O_5_ was dispersed into 36 mL H_2_O under vigorous magnetic stirring at room temperature. Then, 1 mL of H_2_O_2_ (30 wt.%) was slowly dropped into the above solution to obtain a clear orange-red solution. After that, 20 mmol ZnSO_4_ and 3.6 mL GO solution (10%) were added, in order. The obtained suspension was further vigorously stirred for 0.5 h. Last, the mixed suspension was sealed in a 50 mL Teflon-lined stainless-steel autoclave and maintained at 120 °C for 6 h. After reaction, it was cooled to the room temperature naturally, and dark-green precipitates were obtained by suction filtration and washed with H_2_O and freeze-dried for 48 h. For comparison, ZnVOH was prepared without GO solution.

The composition and structure of ZnVOH/rGO were characterized by X-ray diffraction (XRD), Fourier transform infrared spectroscopy (FTIR), Raman spectroscopy, Energy-dispersive X-ray spectrometer (EDS) elemental mapping and inductive coupled plasma emission spectrometer (ICP). Morphology of ZnVOH/rGO was characterized by scanning/transmission electron microscopy (SEM/TEM). The details are represented in the Appendix A.

The cathode was prepared by mixing 80 wt% ZnVOH/rGO, 10 wt% carbon black and 10 wt% polyvinylidene with the N-methyl-2-pyrrolidone as solvent. The slurry was coated onto the above slurry on a clean circular Ti foil (12 mm diameter). After the cathode was dried at 60 ℃ overnight, the 2032-typed AZIB were assembled with glass fiber as the separator, a circular Zn plate of the same dimensions as the anode and 3 mol L^−1^ [Zn(CF_3_SO_3_)_2_] as the electrolyte. The mass loading of the ZnVOH/rGO is about 2 mg. The electrochemical properties of Zn//ZnVOH/rGO batteries were characterized by cyclic voltammetry (CV), electrochemical impedance spectroscopy (EIS), galvanostatic/intermittent titration technique (GITT) and galvanostatic charge–discharge (GCD) in a potential range of 0.2~1.4 V (vs. Zn^2+^/Zn).

## 4. Conclusions

In summary, ZnVOH/rGO was successfully prepared by a facile hydrothermal route. The recombination with rGO can boost the Zn^2+^ storage of Zn^2+^-pre-intercalated VOH. ZnVOH/rGO exhibits much higher specific capacity, better cycle stability and rate performance than ZnVOH. The achieved performances of ZnVOH/rGO even exceed that of some state-of-the-art cathodes for AZIBs. This is mainly due to: (a) Zn^2+^ in a “pillar” structure expanding interlayer distance of VOH enhances efficient Zn^2+^ migration, (b) rGO improves the fast electron flow and (c) both materials stabilize the structure.

## Figures and Tables

**Figure 1 molecules-27-05387-f001:**
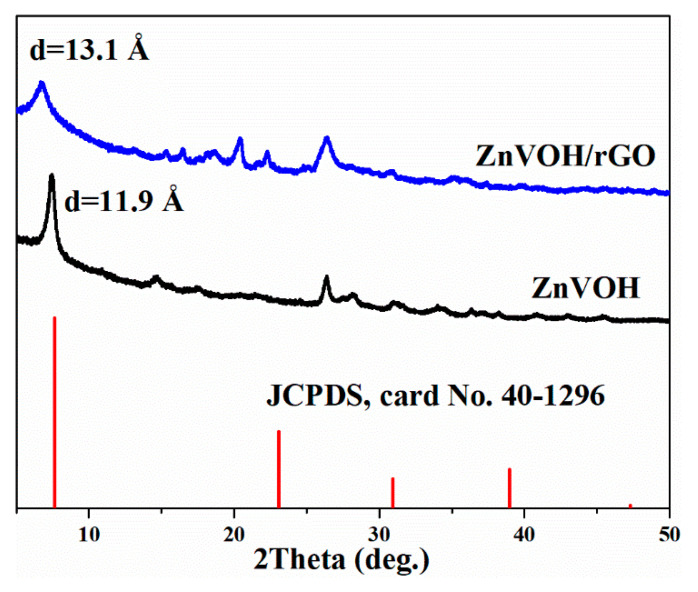
XRD patterns of ZnVOH and ZnVOH/rGO.

**Figure 2 molecules-27-05387-f002:**
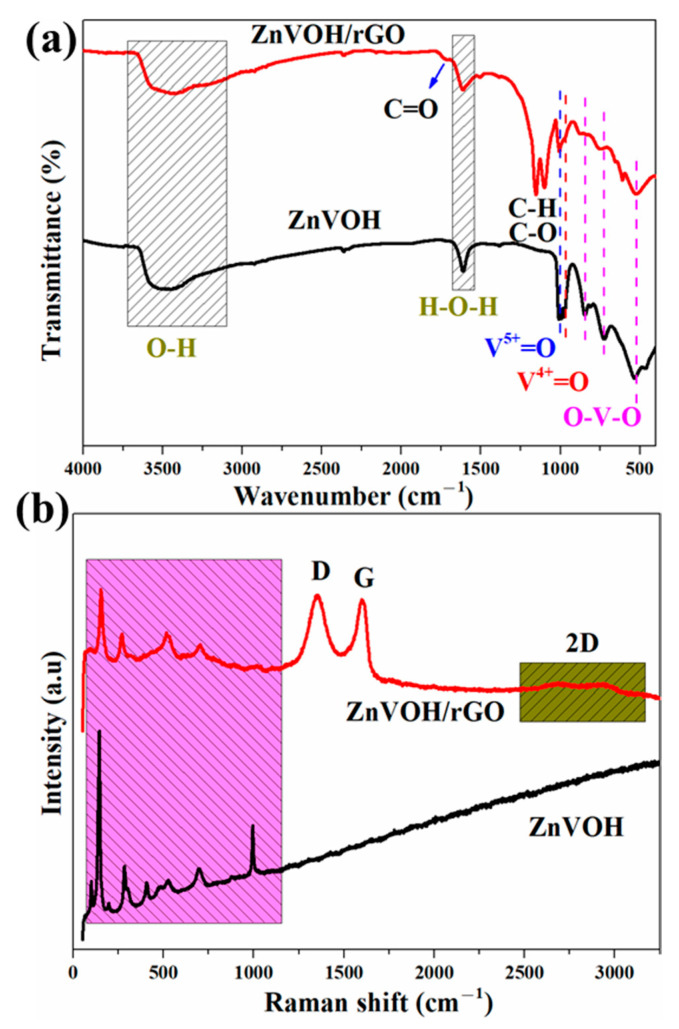
FTIR (**a**) and Raman (**b**) spectra of ZnVOH and ZnVOH/rGO.

**Figure 3 molecules-27-05387-f003:**
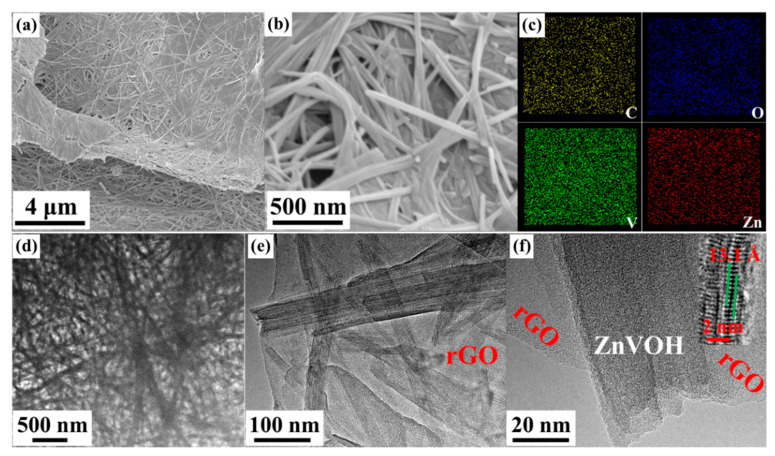
Morphologies of ZnVOH/rGO: (**a**,**b**) SEM images; (**c**) elemental mapping images collected form Appendix A; (**d**–**f**) TEM images and a HRTEM image is inserted.

**Figure 4 molecules-27-05387-f004:**
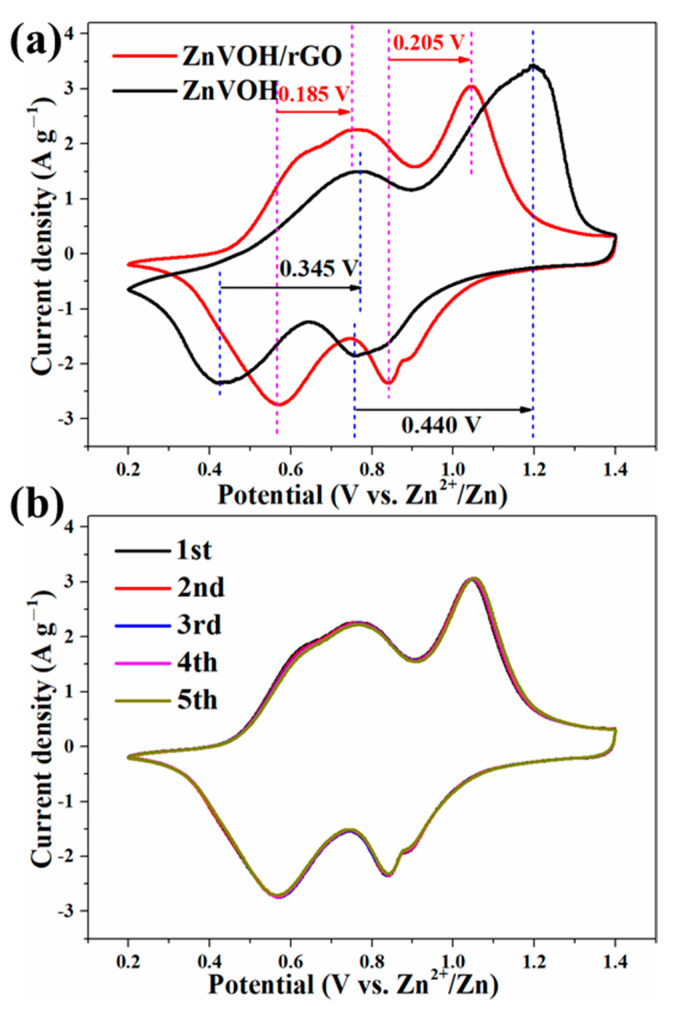
(**a**) Comparative CV curves of ZnVOH and ZnVOH/rGO; (**b**) CV curves of ZnVOH/rGO at the first 5 cycles.

**Figure 5 molecules-27-05387-f005:**
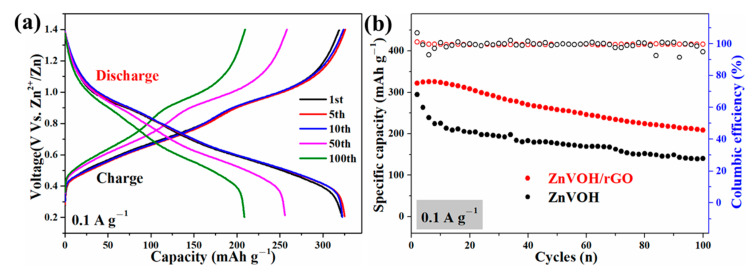
(**a**) GCD curves of ZnVOH and ZnVOH/rGO; (**b**) cycle performance of ZnVOH and ZnVOH/rGO at 0.1 A·g^−1^.

**Figure 6 molecules-27-05387-f006:**
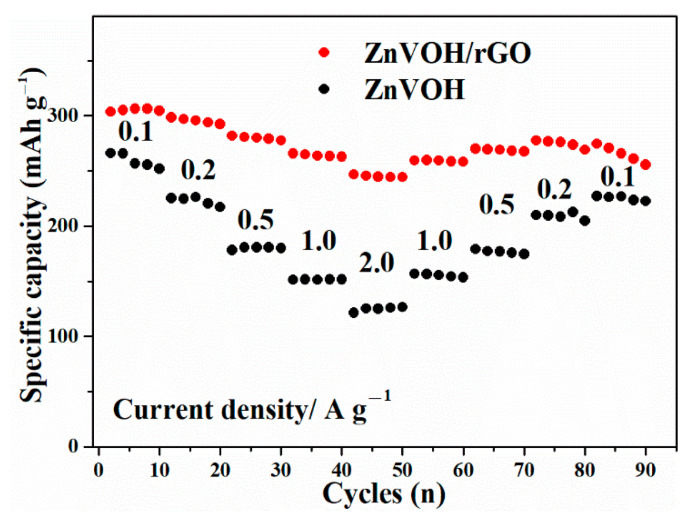
The rate performances of ZnVOH and ZnVOH/rGO.

**Figure 7 molecules-27-05387-f007:**
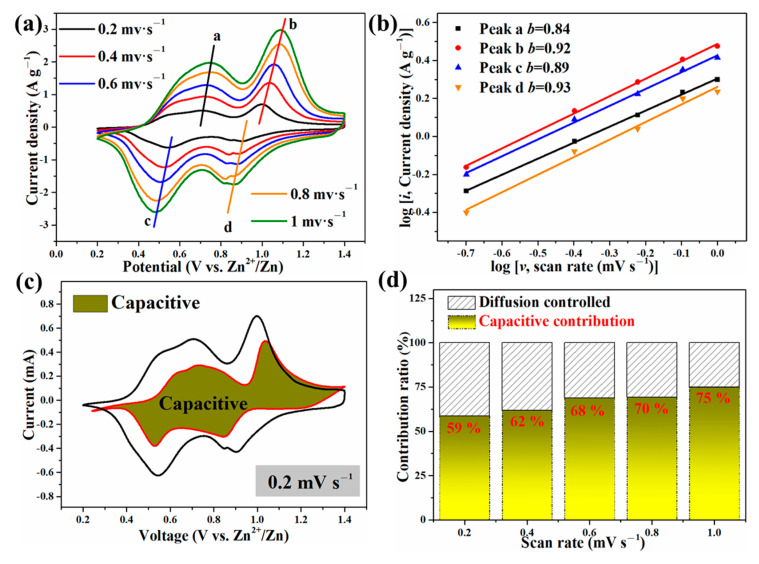
The electrochemical kinetics of Zn//ZnVOH/rGO battery: (**a**) CV curves at 0.2~1 mV·s^−1^, (**b**) The relationship between log (sweep rate) and log (peak current); (**c**) A typical CV curve with capacitive contribution; (**d**) capacitive contributions at current densities.

## Data Availability

No information.

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
