# Peer review of "Synthesis of Zn2+-Pre-Intercalated V2O5·nH2O/rGO Composite with Boosted Electrochemical Properties for Aqueous Zn-Ion Batteries"

_molecules, 2022, doi:10.3390/molecules27175387_

Round 1

Reviewer 1 Report

Review report

The authors have reported on enhanced Zn-ion storage material made up of a composite of Zn2+ intercalated VOH integrated with rGO.  This obviously opens a new dimension for the development of electrode materials for Zn-ion batteries and could also be developed for some electrochemical storage devices.

General comments

Generally, the authors have provided an adequate review of the background with the appropriate references, some additional information and supporting data are required. Also, the introduction, especially the first paragraph, requires heavy editing to improve clarity and eliminate ambiguity. Also, spell checks are required. A lot of frivolous terms were utilized. This must be rectified. I made some comments about them in the pdf.

The authors have provided enough data to support their conclusions. However, there are a lot of inconsistencies (e.g. in text, the authors use V-O-V (Line 86), but in Figure 2a this is referred to as O-V-O) and ambiguity and spell-checks are required.

Also, the figures in the main article must be referenced before referencing the supplementary material.

The approach used by the authors to produce the composite materials is simple and offers a broader opportunity to form different complexes for further testing.

All my comments are corrections have been made on the attached PDF’s

Author Response

The authors have reported on enhanced Zn-ion storage material made up of a composite of Zn2+ intercalated VOH integrated with rGO.  This obviously opens a new dimension for the development of electrode materials for Zn-ion batteries and could also be developed for some electrochemical storage devices.

General comments:

  1. Generally, the authors have provided an adequate review of the background with the appropriate references, some additional information and supporting data are required. Also, the introduction, especially the first paragraph, requires heavy editing to improve clarity and eliminate ambiguity. Also, spell checks are required. A lot of frivolous terms were utilized. This must be rectified. I made some comments about them in the pdf.

A: Thank you for your valuable suggestion. According to your suggestion, we carefully checked the manuscript and corrected it with Yellow highlights after careful consideration. For the questions you asked in the pdf, we will also explain them one by one below:

Answers to your suggestions in line 37 of the original article: Thank you for your valuable suggestion. AZIBs has higher safety and environmental friendliness than lithium and other organic electrolyte batteries because it uses water electrolyte. Meanwhile, zinc, a metal that is more abundant and therefore cheaper to make, stands out from a range of batteries. This is explained in more detail in the inserted reference.

Answers to your suggestions in line 98 of the original article: Thank you for your valuable suggestion. The understanding of high defect density here is that compared with other literatures, this value can be considered to represent a high carbon defect density (J. Colloid Interface Sci. 531 (2018) 382–393, Energy Stor. Mater. 17 (2019) 143–150, Energy Stor. Mater. 13 (2018) 168–174).

The answer is incorporated in the Revised Manuscript with Yellow highlight.

  1. The authors have provided enough data to support their conclusions. However, there are a lot of inconsistencies (e.g. in text, the authors use V-O-V (Line 86), but in Figure 2a this is referred to as O-V-O) and ambiguity and spell-checks are required.

A: Thank you for your valuable suggestion. After your reminder, we checked the manuscript carefully, found some inconsistencies words, and corrected them with Yellow highlights after meticulous consideration.

The answer is incorporated in the Revised Manuscript with Yellow highlight.

Also, the figures in the main article must be referenced before referencing the supplementary material.

A: Thank you for your valuable suggestion. The figures in the main article and supplementary material are referenced in the order they appear.

The approach used by the authors to produce the composite materials is simple and offers a broader opportunity to form different complexes for further testing.

All my comments are corrections have been made on the attached PDF’s

A: Thank you for your valuable suggestion. The details of all comments are revised according to your suggestion in the revised manuscript.

The answer is incorporated in the Revised Manuscript with Yellow highlight.

Reviewer 2 Report

This paper presents interesting data about the electrochemical behaviour  of a Zn2+ preintercalated vanadium oxide / rGO composite. The results seem to show an increase of the capacity in the composite with regard to simple vanadium oxide as reference. Nevertheless, the comparison conditions of the 2  materials are not clear for me : what is the real total Carbon / active material ratio in the 2 electrodes ? In my opinion, the increase of capacity in the composite electrode will be effective only if the ratio is the same in the 2 electrodes. These data are lacking. In other terms, the behaviour of the electrode with composite material must be compared with an electrode containing a mixture of ZnVOH with the same amount of C (carbon black + C coming from RGO).

I suggest this paper can be published in this journal, once the important point above has been clarified and the following questions have been addressed or completed.

·       Line 64. The authors should mention that the capacity of their material is not higher than that reported by Nazar, all the more as that the cycling was in this case at a very high rate (50 mA.g-1 for Nazar’s groupe against 0.1 A.g-1 for the present group).

·       Figure 1. What do the red lines correspond to ? Which material ? Why do they not match with experimental peaks ?

·       The characterization of the composition of the materials is weak. The Zn/V ratio on one hand as well as the C/V ratio on the other hand must be measured and indicated in the text. In particular, it could be interesting to compare the Zn/V ratio in ZnVOH and ZnVOH/RGO and to correlate it with the interlamellar distance. Techniques such as EDS/EDX or ICP titration after mineralization are suggested to determine the composition.

·       Figure 7 and correspond text. The capacitive contribution within the electrodes is quite huge even at low scan rate, which questions the ability of the material with faradaic behaviour, as expected for a good battery material. In addition, the mass loading of material in the electrode is very weak (2mg) and does not correspond to the standards required for a battery material involving reactions in the bulk (at least 10 mg.cm-2).
